# Metal-Bound Methisazone; Novel Drugs Targeting Prophylaxis and Treatment of SARS-CoV-2, a Molecular Docking Study

**DOI:** 10.3390/ijms22062977

**Published:** 2021-03-15

**Authors:** Ahmed Abdelaal Ahmed Mahmoud M. Alkhatip, Michail Georgakis, Lucio R. Montero Valenzuela, Mohamed Hamza, Ehab Farag, Jaqui Hodgkinson, Hisham Hosny, Ahmed M. Kamal, Mohamed Wagih, Amr Naguib, Hany Yassin, Haytham Algameel, Mohamed Elayashy, Mohamed Abdelhaq, Mohamed I. Younis, Hassan Mohamed, Mohammed Abdulshafi, Mohamed A. Elramely

**Affiliations:** 1Faculty of Medicine, Beni-Suef University, Beni Suef 62511, Egypt; Ehab.yaseeen@med.bsu.edu.eg; 2Birmingham Children’s Hospital, Birmingham B4 6NH, UK; 3Sinodos Chemistry Company (SinodosChemistry.com), Tenedou 16 str, 54453 Thessaloniki, Greece; MGeorgakis@SinodosChemistry.com; 4Instituto de Biotecnología, UNAM (ibt.unam.mx), 62210 Cuernavaca, Mexico; lucioric@ibt.unam.mx; 5Faculty of Medicine, Cairo University, Giza 11562, Egypt; mkhamza@kasralainy.edu.eg (M.H.); hishamhosny@kasralainy.edu.eg (H.H.); ahmedmostafakamal@yahoo.com (A.M.K.); mohamedwagih@kasralainy.edu.eg (M.W.); kazamora1979@hotmail.com (A.N.); mohamedelayashy@kasralainy.edu.eg (M.E.); mohamedabdelhaq76@hotmail.com (M.A.); drhassanmohamed@yahoo.com (H.M.); 6Jaqui Hodgkinson Communications, 3722JK Bilthoven, The Netherlands; info@jaqui.nl; 7Essex Cardiothoracic Centre, MSE Foundation Trust, Basildon SS16 5NL, UK; 8Faculty of Medicine, Fayoum University, Faiyum 63514, Egypt; hmy00@fayoum.edu.eg; 9Aberdeen Royal Infirmary Hospital, Aberdeen AB25 2ZN, UK; hzien2002@yahoo.com; 10Cambridge University Hospital, Cambridge CB2 0QQ, UK; ismaiel.m@gmail.com; 11Leeds General Infirmary Hospital, Leeds LS1 3EX, UK; 12Imam Abdulrahman Bin Faisal University Hospital, University of Dammam, Dammam 34221, Saudi Arabia; mohammedabdulshafi@yahoo.com; 13National Cancer Institute, Cairo University, Giza 11796, Egypt; mramely@hotmail.com

**Keywords:** COVID-19, SARS-CoV-2, molecular docking, treatment, prophylaxis

## Abstract

SARS-CoV-2 currently lacks effective first-line drug treatment. We present promising data from in silico docking studies of new Methisazone compounds (modified with calcium, Ca; iron, Fe; magnesium, Mg; manganese, Mn; or zinc, Zn) designed to bind more strongly to key proteins involved in replication of SARS-CoV-2. In this in silico molecular docking study, we investigated the inhibiting role of Methisazone and the modified drugs against SARS-CoV-2 proteins: ribonucleic acid (RNA)-dependent RNA polymerase (RdRp), spike protein, papain-like protease (PlPr), and main protease (MPro). We found that the highest binding interactions were found with the spike protein (6VYB), with the highest overall binding being observed with Mn-bound Methisazone at −8.3 kcal/mol, followed by Zn and Ca at −8.0 kcal/mol, and Fe and Mg at −7.9 kcal/mol. We also found that the metal-modified Methisazone had higher affinity for PlPr and MPro. In addition, we identified multiple binding pockets that could be singly or multiply occupied on all proteins tested. The best binding energy was with Mn–Methisazone versus spike protein, and the largest cumulative increases in binding energies were found with PlPr. We suggest that further studies are warranted to identify whether these compounds may be effective for treatment and/or prophylaxis.

## 1. Introduction

Severe acute respiratory coronavirus disease 2019 (COVID-19), caused by novel SARS-CoV-2, was first documented in December 2019 [1]. The overall mortality rate is currently estimated to be 2.2% and varies across regions [2]. SARS-CoV-2 has proven to be more hazardous than previous viral outbreaks: the H1N1 pandemic caused 12,429 deaths worldwide over a year, while SARS-CoV-2 has caused more than 365,886 deaths in the USA alone [3]. SARS-CoV-2 also has a higher transmission rate, with an estimated basic reproductive number (BRN) of between 2 and 4.5, compared to the BRN of 1.8 in the 1918 influenza pandemic, which resulted in 50 million deaths [4,5]. Few drugs have so far been shown to be effective against COVID-19 [6,7,8], and even with the recent advent of vaccines [9,10], there remains an urgent need for extra pharmacological treatments, particularly ones that can be manufactured at scale and used safely in the early stages of the disease or as post-exposure prophylaxis.

Viral attack of human cells occurs through a sequence of steps: SARS-CoV-2 enters the human body through angiotensin-converting enzyme-II (ACE-II) receptors, found in the lungs, heart, kidneys, and gastrointestinal tract [11]. SARS-CoV-2 enters human cells via the spike protein binding to the ACE-II receptor [12], making the spike protein a target of interest for drug screening [13], although the newest, potentially more virulent strain of SARS-CoV-2, identified in December 2020 in the UK, along with other recent new variants, have spike protein mutations [11,14].

In this study, we chose to focus on molecular docking studies as they are one of the most frequently used tools in structure-based drug design, allowing for characterization of binding behaviors of drug molecules in relation to therapeutic targets [15,16]. SARS-CoV-2 spike protein was chosen as a target in this docking study, as in others, because of its key role in mediating viral entry to human cells [13]. Other potential targets screened were two proteases essential for viral protein processing (main protease (MPro) and papain-like protease (PlPr)) and RNA-dependent RNA polymerase (RdRp), which is essential for replication [17].

Previous work by Shah and coworkers involved a docking study of SARS-CoV-2 on 61 molecules with known antiviral activities, in which Methisazone had marked binding interactions with SARS-CoV-2 enzymes. Methisazone was shown to interact with 5R7Z, 5R80, and 5R81 enzymes with dock score values of −7.542, −6.829, and −6.928, respectively [18]. Methisazone, a thiosemicarbazone, inhibits synthesis of structural viral proteins and interrupts viral assembly in pox viruses [19]. Use for smallpox prophylaxis was well tolerated, with the only adverse events reported being nausea and vomiting [20,21,22]. Thiosemicarbazones have been previously reported to have enhanced activity when metal-chelated [23]. In the present study, we aimed to use molecular docking not to screen for possible drug molecules, but to confirm our hypotheses that modification of the previously identified drug candidate, Methisazone, could increase its ability to bind to SARS-CoV-2 proteins. Thus, we aimed to conduct a molecular docking study to test the role of modified calcium (Ca), iron (Fe), manganese (Mn), magnesium (Mg) and zinc (Zn) Methisazone against SARS-CoV-2.

## 2. Results

### 2.1. Molecular Docking Interaction Data

Interaction data for free Methisazone and the five metal complexes we selected, with four protein targets from SARS-CoV-2, are shown in Table 1. The binding energy differences produced with the insertion of metals are presented in Table 2. The 6M71_A protein binding was minimally influenced by metal addition: in some cases, the interaction energies modeled were weaker with the metal ligands tested. The lowest binding energy as an absolute value (weaker binding) for Methisazone was found with 6Y2E_A and 6W9C. Ligands targeting 6W9C led to the greatest change in binding energy, with a total of −1.8 kcal/mol as a cumulative increase for all metal complexes; the average increase of the binding energy of metal complexes was −0.36 kcal/mol. The 6VYB and 6Y2E_A proteins showed cumulative increases of −1.6 and −1.4 kcal/mol for five metal complexes, which translated to an average increase of affinity by −0.32 and −0.28, respectively. Modification of Methisazone with Fe and Zn led to the greatest changes in binding energy of −1.2 and −1.1 kcal/mol for the three proteins, respectively. The respective average increases per protein were −0.3 and −0.275 kcal/mol. The other three metals followed closely with Mn, Mg, and Ca having −1.0, −0.9, and −0.6 kcal/mol cumulative increases for the three proteins (average increase of −0.25, −0.225, and −0.15 kcal/mol, respectively).

The highest binding interactions were found with 6VYB, with the highest being observed with Mn chelation of Methisazone, followed by Zn and Ca, and Fe and Mg. The highest increases of binding energy were observed with Mn and for the 6VYB case, Fe for 6W9C, Mg for 6W9C, and Zn for 6Y2E_A.

### 2.2. Docking Visualization

Figure 1, Figure 2, Figure 3, Figure 4 and Figure 5 visualize the docking sites in the four SARS-CoV-19 proteins tested. For better visual clarity, all labels have been removed from the images. 6Y2E_A had two binding pockets: one multi-occupant site and three sites with single occupants (Figure 1).

As shown in Figure 2, there are three different absorption sites for the various metal complexes in 6W9C. The graphical representation is in agreement with the calculated binding energies. The three different sites/pockets appear to have increasing stereochemical obstacles that can only be overcome by the increased affinities of the ligand and specific amino acids in the pockets.

Similar results were obtained with 6VYB (Figure 3): three docking sites were found, and the highest interaction energy of −8.3 kcal/mol was observed at the site bound only by the Mn—Methisazone complex. The rest of the high-affinity interactions (−7.9 and −8.0 kcal/mol, depending on the metal) were grouped at a second docking site. A third docking site was found which also bound all five metal complexes, with slightly different orientations.

6Y2E_A yielded five docking sites, with a narrow docking energy range (from −6.2 to −6.7 kcal/mol, Figure 4). A dispersion of binding sites was observed rather than focused, high-energy interactions.

Figure 5 depicts the strongest interaction of Ca–Methisazone with 6M71_A; two major interactions are evident, between the complex and THR261 and SER260. The close environment of the pocket (ARG291, VAL257, PRO403, and ASN570) could also be targeted in a small-scale investigation with a modified ligand.

Finally, Figure 6 shows the highest binding interaction found in this study: Mn–Methisazone in a 6VYB pocket. The interactions were between the ligand and ARG2767, VAL2768, SER849, GLU850, and CYS851. The pockets in 6VYB appeared to be more ”confined” in comparison to the binding pockets in 6M71_A, which may have been the cause of the stronger binding interaction found.

## 3. Discussion

We investigated in silico the inhibiting role of Methisazone and that of the modified drugs Ca–, Fe–, Mg–, Mn–, and Zn–Methisazone against SARS-CoV-2 proteins. We demonstrated that these metal-modified drugs display better binding energies for some SARS-CoV-2 proteins, suggesting that these compounds may provide effective treatment and/or prophylaxis. We detected a range of increased binding with metal-modified Methisazone, with the highest binding affinities being found for the spike protein 6VYB and Mn–Methisazone. In agreement with previously published work, free Methisazone was found to interact with more than two of the protein structures we tested from SARS-CoV-2 [20], which we also found with the Methisazone–metal complexes. It can be claimed, based on previously published molecular docking studies, that a threshold of −6.0 to −6.5 kcal/mol can be used for “binding” or ”non-binding” classifications [18,24]. Docking energies in a range up to roughly 50 kcal/mol indicate a weak physical interaction between the ligand and the protein/enzyme and not a chemical interaction [20]. According to recently published comparative work, only binding energies −7.9 and below should be considered as strong interacting compounds/complexes. In a similar study, the top hits were predicted to be as low as −8.7 kcal/mol of binding energy for approved drugs. Experimental drugs exhibited binding energies as low as −9.1 kcal/mol [25].

Visualization of preferred binding sites and pockets assisted in better understanding of the combined interactions of multiple protein segments and of the possible existence of non-specific interactions. The narrow binding profiles that were observed visually confirmed that non-specific interactions should be negligible in metal–Methisazone docking in the four proteins studied.

The initial optimization of Methisazone and Methisazone–metal complexes with a “near-infinite basis set” allowed for the accurate description of the Methisazone–metal complexing and the prediction of the stability of the complex. Very weak interactions at that complexing stage could be the result of either an inappropriate investigation approach or the real absence of interactions between Methisazone and metals. This would lead to complexes not being stable enough to be used as potential drugs. Our approach, based on the employment of a “near-infinite basis set” and combined geometry optimization–molecular dynamics, confirmed the sufficient binding strength of Methisazone with the metals investigated. The process that was followed can be described as follows:-Methisazone and the metal were placed in random positions (at least 20 different ones).-A geometry optimization reached a local energetic minimum.-A molecular dynamics simulation was employed to move the Methisazone–metal system from equilibrium.-If the attempt was not successful, the local minimum was the global minimum, and the complex represented the real structure. All properties of the complex were fed to Chimera for the molecular docking investigation (end of optimization).-If the attempt was successful, a new geometry optimization was run to find a new energetic minimum.-The process was continued until the global minimum was reached, and all properties were fed to Chimera for the molecular docking investigation (end of optimization).

Overall, the PlPr (6W9C) led to the greatest cumulative change in binding energy with the various metal ligands tested, and this protein may thus be a useful target in further drug development. Our study showed that metal binding led to increased affinity between the ligand and the target proteins, and we hypothesize that a combined metal complex could lead to even higher affinity towards the binding pockets we identified in the four SARS-CoV-2 proteins tested. A simultaneous docking or molecular dynamics study of more than one type of metal complex in the identified binding pockets may increase the chances of reaching higher docking energies.

## 4. Materials and Methods

### 4.1. Structure Preparation

The structure of Methisazone was prepared in HyperChem 8.0 (HyperCube Inc., Gainesville, FL, USA). The metal complexes of Methisazone were also prepared in the HyperChem 8.0 environment since HyperChem offers a more powerful environment for the optimization of small molecular systems via ab initio and density functional theory calculations than Chimera. Free Methisazone was optimized using ab initio theory at the 6–31G* level on HyperChem, with partial charges retained for the second (complexation) step. The 6–31G* level offered the richest basis set to model the interactions of the metal ions with Methisazone that our computational power could handle. The structures of SARS-CoV-2 proteins were downloaded from the Protein Data Bank (RCSB.org, accessed on 15/07/2020). Next, SARS-CoV-2 protein structure models were compiled for 6M71_A (SARS-Cov-2 RdRp chain A, PDB ID: 6M71); 6VYB (SARS-CoV-2 spike ectodomain structure, with the N-acetyl glucosamine removed, PDB ID: 6VYB); 6W9C (PlPr of SARS-CoV-2, PDB ID: 6W9C), and 6Y2E_A (SARS-CoV-2 MPro chain A, unmodified, PDB ID: 6Y2E). Missing protons were added where required, and energetic- based stability choice of multiple locations for atoms was employed. Charges were calculated based on the AMBER FF14SB force field for all protein structures. This editing was performed using the UCSF Chimera software, version 1.14 (Resource for Biocomputing, Visualization, and Informatics, San Francisco, CA, USA), generating mol2 files that were used by the AutoDockTools software (v4.2, The Scripps Research Institute, San Diego, CA, USA) to generate pdbqt files for AutoDock Vina, preserving the charges found in the mol2 files (through assigning of AutoDock 4 atom types and merging of non-polar hydrogen atoms).

The second step of our approach involved a mixed molecular dynamics and geometry optimization process for all Methisazone–metal complexes of interest. These two methods were combined to optimize analysis of the entire molecular space, without trap creation. A molecular dynamics substep was utilized after each successful geometry optimization, to enable confirmation of a global energetic minimum or at least the deepest local minima for the complex. The molecular dynamics substep was run under constant temperature (310K), time step of 0.001 ps (to avoid “breaking” of bonds due to bond oscillation frequencies), for a total time of 5 ps. Continuous kinetic energy, temperature and total energy monitoring was applied. The same process was followed for the preparation of all Methisazone–metal complexes: Ca, Fe, Mg, Mn, and Zn. Figure 7 displays three of the metal complexes created in HyperChem. All files generated were initially saved including partial charges, and then AutoDockTools was used to create PDBQT files for the molecular docking simulations. The structures created, along with the metal–Methisazone distances suggested that the complexing occurred via electrostatic and dispersion forces (distances 1.5–1.9 times the lengths of the respective covalent bonds). Thus, it was concluded that Methisazone–metal complexes are stable and can be delivered at their target.

### 4.2. Docking Analysis

The available crystallographic structures for four SARS-CoV2 proteins were downloaded from Protein Data Bank (RCSB.ORG) and used in Chimera software to create the protein structures of the four proteins of interest. The Lamarckian genetic algorithm 24 was employed, using 20 evaluations for run and 25 million iterations for result generation, with all other parameters set to their default values. The interaction potential grids for each atom in the ligands to be screened were calculated using the iterative local search method of AutoDock Vina. Spacing was chosen and adjusted as follows: the investigation was initiated at a low-resolution mode, at grid spacing of 0.8 Angstroms to reach a first estimate. Then, a high-resolution grid was employed with 0.375 Angstroms spacing for more accurate results. Different potential grids were prepared for each protein to create an overlapping space for Methisazone and its metal complexes, which was then used to model interactions with the protein structures. Overlapping potential grid segment positions were calculated using a Python script to create input files for AutoGrid and for AutoDock, submit the AutoGrid and AutoDock runs, and validate all runs. An additional Python script was employed to link the docking results for the same protein and the same ligand, and to compare results across proteins and among different ligands.

### 4.3. Drug Candidate Analysis

The results per candidate complex were evaluated. AutoDock software reported the binding energies for each Methisazone–protein pair in kcal/mol. Binding energies between the proteins and complexes were calculated as the difference between the final docked structure (protein–ligand) and the initial energies of the protein and Methisazone–metal complex.

## 5. Conclusions

The identification of SAR-CoV-2 protein binding pockets in this study could form the basis of focused investigations utilizing ab initio or density functional theory in a system of 200–500 atoms per pocket. Such investigations would further clarify the atomic interactions between the ligand and the target amino acids, thus providing additional information for the preparation of the most efficient COVID-19 drugs. Lastly, we believe that Mn–Methisazone should be studied in both in vitro and in vivo studies, not only due to its effectiveness in this docking study, but also because much is already known about the “mother” drug Methisazone and its use in oral form. We believe an oral drug will be needed for widespread prophylaxis against COVID-19 syndrome.

## 6. Patents

International application No. PCT/IB2020/055863.

## Figures and Tables

**Figure 1 ijms-22-02977-f001:**
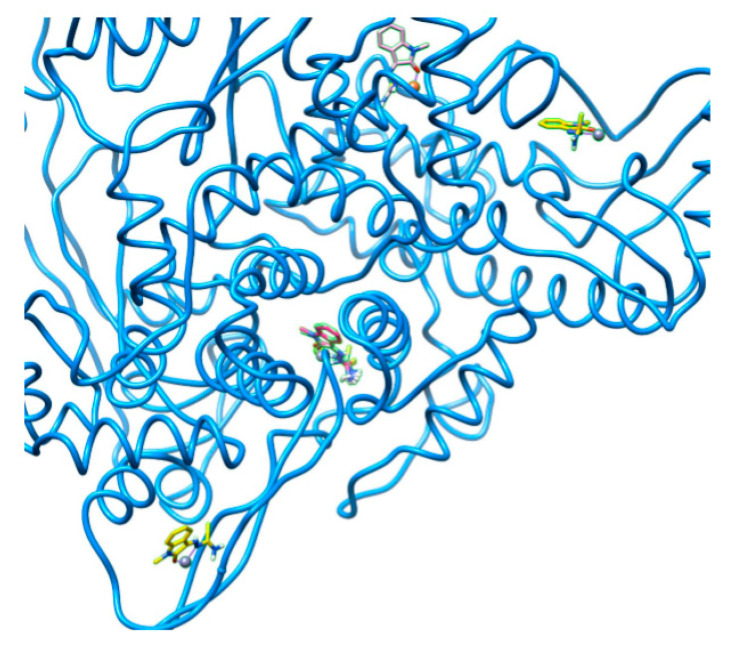
Visualization of low-energy docking sites in 6M71_A using Chimera software. Two binding pockets can be observed: one multi-occupant site and three sites with single occupants. Molecular graphics performed with UCSF Chimera, developed by the Resource for Biocomputing, Visualization, and Informatics at the University of California, San Francisco, with support from NIH P41-GM103311.

**Figure 2 ijms-22-02977-f002:**
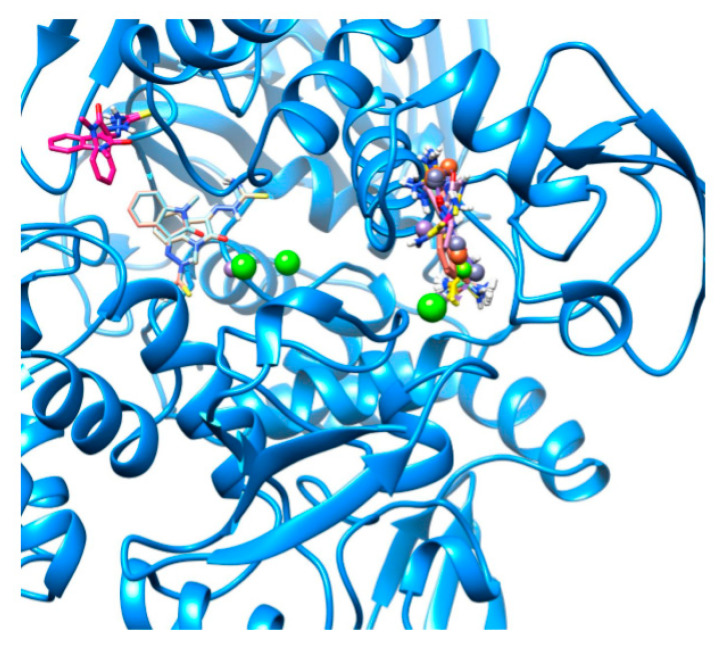
Visualization of high-energy docking sites in 6W9C. Three different absorption sites are evident. Free Methisazone and its metal complexes appear to dock in different pockets. The most populated site accommodates all metal complexes, the second most populated holds two metal complexes, and the third is occupied only by free Methisazone. Molecular graphics and analyses performed with UCSF Chimera, developed by the Resource for Biocomputing, Visualization, and Informatics at the University of California, San Francisco, with support from NIH P41-GM103311.

**Figure 3 ijms-22-02977-f003:**
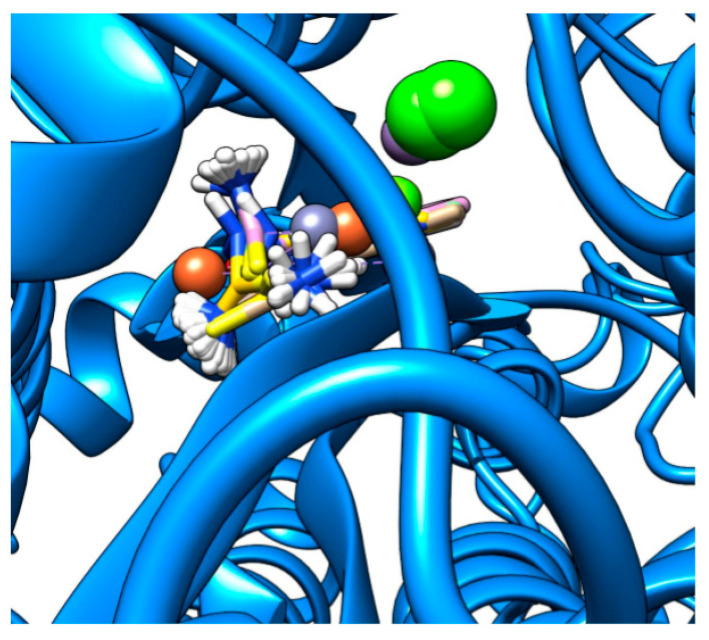
Visualization of three docking sites found in 6VYB. The highest interaction energy was observed at the site bound only by the Mn–Methisazone complex. Molecular graphics and analyses performed with UCSF Chimera, developed by the Resource for Biocomputing, Visualization, and Informatics at the University of California, San Francisco, with support from NIH P41-GM103311.

**Figure 4 ijms-22-02977-f004:**
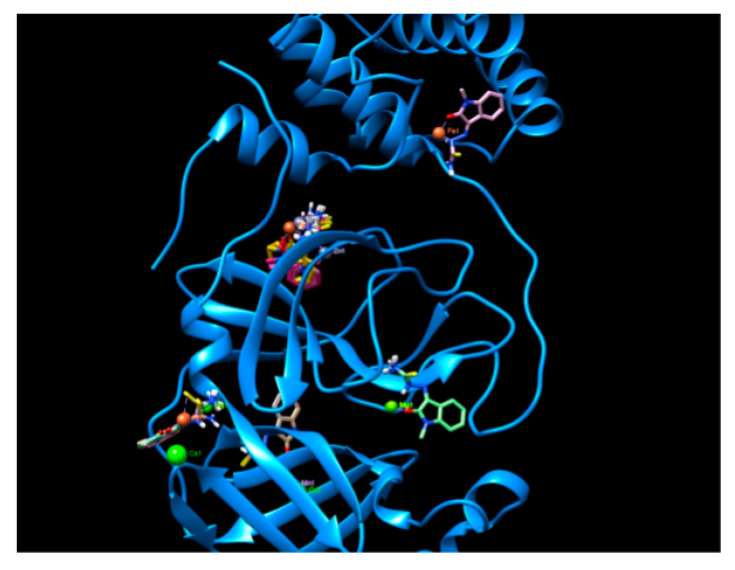
Visualization of five docking sites in 6Y2E_A. Two docking sites have single occupancy (Fe and Mg), one has two occupants (Ca and Mn) and the other two can be occupied by all metal complexes. Molecular graphics and analyses performed with UCSF Chimera, developed by the Resource for Biocomputing, Visualization, and Informatics at the University of California, San Francisco, with support from NIH P41-GM103311.

**Figure 5 ijms-22-02977-f005:**
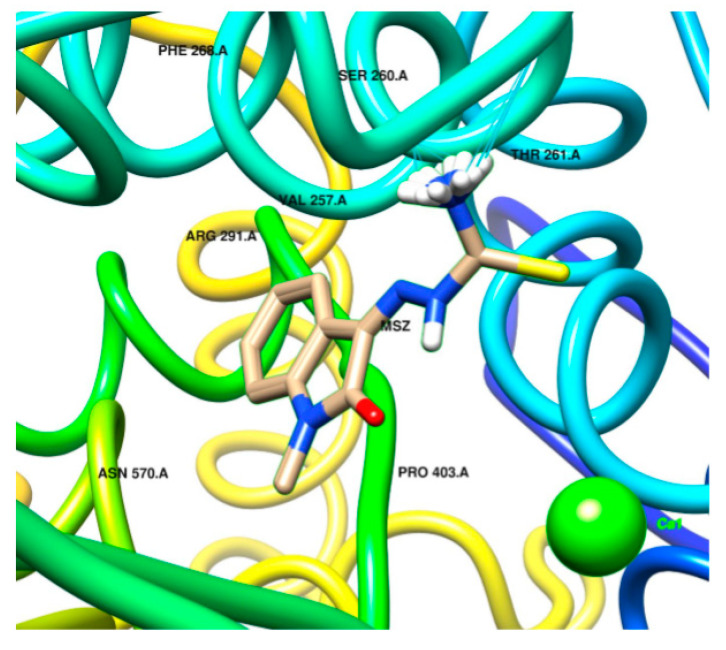
Visualization of docking of Ca–MSZ in 6M71_A. Molecular graphics and analyses performed with UCSF Chimera, developed by the Resource for Biocomputing, Visualization, and Informatics at the University of California, San Francisco, with support from NIH P41-GM103311.

**Figure 6 ijms-22-02977-f006:**
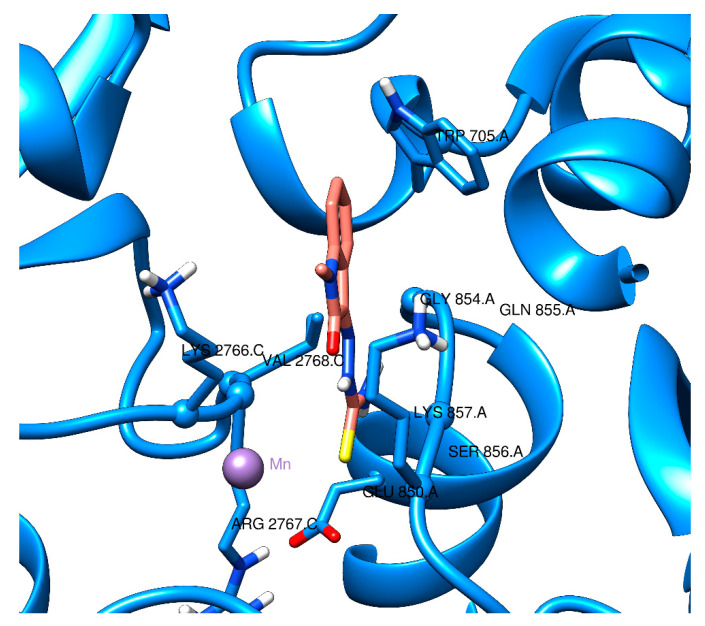
Visualization of docking of Mn–MSZ in 6VYB. Molecular graphics and analyses performed with UCSF Chimera, developed by the Resource for Biocomputing, Visualization, and Informatics at the University of California, San Francisco, with support from NIH P41-GM103311.

**Figure 7 ijms-22-02977-f007:**
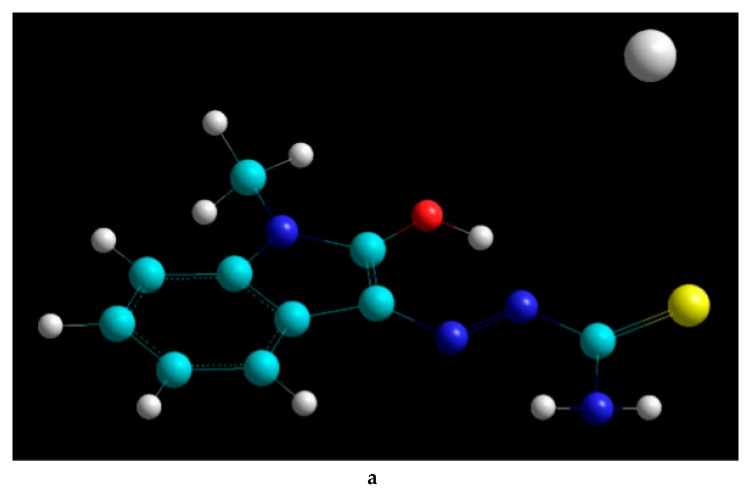
HyperChem representation of (**a**) Methisazone–Ca, (**b**) Methisazone–Zn, and (**c**) Methisazone–Mn. Molecular graphics and analyses performed with HyperChem, HyperCube Inc, USA. Color interpretation in the Figure: Light Blue/ Cyan for Carbon, Deep Blue for Nitrogen, White for Hydrogen Red for Oxygen, Yellow for Sulfur, White (larger atom) for Metals.

**Table 1 ijms-22-02977-t001:** Binding energies of Methisazone and its metal complexes for four SARS-CoV-2 proteins (best results) (kcal/mol).

Complex	6M71_A PDB: 6M71	6VYB PDB: 6VYB	6W9C PDB: 6W9C	6Y2E_A PDB: 6Y2E
MSZ	−7.1	−7.7	−6.8	−6.2
Ca–MSZ	−7.1	−8	−6.9	−6.4
Fe–MSZ	−6.9	−7.9	−7.4	−6.6
Mg–MSZ	−6.8	−7.9	−7.3	−6.4
Mn–MSZ	−7.1	−8.3	−7.1	−6.3
Zn–MSZ	−7	−8	−7.1	−6.7

6M71_A, SARS-Cov-2 RNA-dependent RNA polymerase chain A; 6VYB, SARS-CoV-2 spike ectodomain structure; 6W9C, papain-like protease of SARS-CoV-2; 6Y2E_A, SARS-CoV-2 main protease chain A; PDB, Protein Data Bank; MSZ, Methisazone.

**Table 2 ijms-22-02977-t002:** Binding energy changes with respect to Methisazone complexation (kcal/mol).

Complex	6M71_A	6VYB	6W9C	6Y2E_A
Ca–MSZ	0	−0.3	−0.1	−0.2
Fe–MSZ	0.2	−0.2	−0.6	−0.4
Mg–MSZ	0.3	−0.2	−0.5	−0.2
Mn–MSZ	0	−0.6	−0.3	−0.1
Zn–MSZ	0.1	−0.3	−0.3	−0.5

6M71_A, SARS-Cov-2 RNA-dependent RNA polymerase chain A; 6VYB, SARS-CoV-2 spike ectodomain structure; 6W9C, papain-like protease of SARS-CoV-2; 6Y2E_A, SARS-CoV-2 main protease chain A; MSZ, Methisazone.

## Data Availability

Computational Chemistry calculations performed in HyperChem, HyperCube Inc [USA] and molecular dockings performed in Chimera Resource for Biocomputing, Visualization, and Informatics, San Francisco, USA) are available.

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
