# Peer review of "Metal-Bound Methisazone; Novel Drugs Targeting Prophylaxis and Treatment of SARS-CoV-2, a Molecular Docking Study"

_ijms, 2021, doi:10.3390/ijms22062977_

Round 1

Reviewer 1 Report

The manuscript titled "Metal-bound Methisazone; Novel Drugs Targeting Prophylaxis and Treatment of SARS-CoV-2, a Molecular Docking Study" is an interesting mix of techniques used to evaluate the binding mode and theoretical binding affinity of Methisazone derivatives on SARS-CoV-2 enzymes. Nonetheless, authors should better desribe some parts:

Methods

  • More details on Ab initio method used should be explained (why using 6-31G* basis set and why this theory have been chosen)
  • How protein structures were prepared?
  • What kind of MD was adopted?

Results and Discussion

  • Figures quality should be improved (e.g. in fig.7 it is quite impossible to read AA residues number)
  • Authors should add some reference about the sentences reported in the discussion (lines 221-228). Where it is written that a threshold of -6.5 kcal/mol is used to discriminate binders from non binders?
  • in line 213 authors state "we demonstrated..." are the computational results supported by experimental evidences?
  • How computational values of previous published works can be considered as threshold for evaluating other new docking protocols? Are the previously reported data about enthalpy or entropy evaluation? Are the values confirmed by experimental data?
  • How much significant are the differences between different binding energies reported in table 1a? Did authors run a statistical significance analysis of these data? if yes, please provide them.

Author Response

Reviewer one: 

The manuscript titled "Metal-bound Methisazone; Novel Drugs Targeting Prophylaxis and Treatment of SARS-CoV-2, a Molecular Docking Study" is an interesting mix of techniques used to evaluate the binding mode and theoretical binding affinity of Methisazone derivatives on SARS-CoV-2 enzymes. Nonetheless, authors should better desribe some parts:

Methods

  • More details on Ab initio method used should be explained (why using 6-31G* basis set and why this theory have been chosen)

Response: 6-31G* basis set offers better and more reliable results in comparison to 6-31G. The 6-31G basis set is the standard, split valence double zeta basis set (describes core and valence orbitals). The choice of 6-31G* aims at two directions; first, a better description of bonding between metal NP and the ligand (via polarization functions addition), and secondly it meets our computational time requirements (before going to the next level, 6-31G** which we cannot handle in computational time terms)

Changed text  as follows:

Line 97:  “6-31G* level offered the richest basis set to model  the interactions of the metal ions with Methisazone that our computational power could handle. SARS-CoV-2 proteins’ structures were downloaded from the Protein Data Bank (RCSB.org).”

Reviewer one: 

  • How protein structures were prepared?

Response: Protein structures were prepared as described in lines 99-110. Initially the crystallographic structures were obtained from PDB (RCSB.org) and then the described action for each protein took place. Text has been updated:

Line 99  “ SARS-CoV-2 proteins’ structures were downloaded from the Protein Data Bank (RCSB.org). Next, SARS-CoV-2 protein structure models were compiled for 6M71_A (A chain only of SARS-Cov-2 RdRp chain A, PDB ID: 6M71); 6VYB (SARS-CoV-2 spike ectodomain structure, with the N-acetyl glucosamine removed, PDB ID: 6VYB); 6W9C (PlPr of SARS-CoV-2, PDB ID: 6W9C) and 6Y2E_A (SARS-CoV-2 MPro chain A, unmodified, PDB ID: 6Y2E). Missing protons were added where required, and energetic- based stability choice of atoms multiple locations was employed. Charges were calculated based on the AMBER FF14SB force field for all protein structures. This editing was performed using the UCSF Chimera software, version 1.14, generating .mol2 files that were used by the AutodockTools software to generate pdbqt files for Autodock vina, preserving the charges found in the mol2 files (through assigning of Autodock 4 atom types and merging of non-polar hydrogen atoms).”

Reviewer one: What kind of MD was adopted?

Response:  Text was added:

Line 116:   “Molecular Dynamics were run under constant temperature (310K), time step of 0.001 ps (to avoid ‘breaking’ of bonds sue to bond oscillation frequencies), for a total time of 5 ps. Continuous kinetic energy, temperature and total energy monitoring was applied.”

For Molecular docking: The Iterative local search method of Autodock Vina was used, using a random numbers seed of 130476, the default number of reported binding poses and an exhaustiveness of the grid box volume times 8/27000, rounded to its next upper integer number. The interaction potential grids for each atom in the ligands to be screened is calculated automatically by Autodock Vina, but the quality of the docking results is inversely proportional to the size of the grid, thus, we used a potential grid that would have been considered as high-resolution in Autodock 4; a grid with spacing 0 0.375Å. Autodock 4 has a maximum number of grid points of 126 points along each dimension, which results in a cube with a side of 46.875Å. Different potential grids were prepared for each protein to cover all the protein volume with an overlapping space of 10.811Å (space found to fit a methisazone molecule and its metal complexes).

For clarity, the following text in red was added to the paper:

Line 140 ”….. The interaction potential grids for each atom in the ligands to be screened was calculated using the iterative local search method of Autodock Vina.”

Reviewer one: 

Results and Discussion

  • Figures quality should be improved (e.g. in fig.7 it is quite impossible to read AA residues number)

Response: Corrected, figure 7 has been updated.

  • Authors should add some reference about the sentences reported in the discussion (lines 221-228). Where it is written that a threshold of -6.5 kcal/mol is used to discriminate binders from non binders?

Response:  binders with energies below -6.5 kcal/ mol are rejected as potential SARS-CoV-2 drugs in the most recent in silico studies, as it is also suggested in the references presented. Taking into consideration the exponential dependence of binding population on the binding energy, this is a suggestion that we followed.  Also, (Shityakov & Förster, 2014) suggest as a limit -6.0 kcal/mol (which we believe that is fairly low)

Text has been changed as follows:

Line 244: It can be claimed based on previous published molecular docking studies, that a threshold of -6.0 to -6.5 kcal/mol is accepted for ‘binding’ and ‘non-binding’ classification (Shityakov & Förster, 2014, Shah et al., 2020).”

Reviewer one: 

  • in line 213 authors state "we demonstrated..." are the computational results supported by experimental evidences?

Response:  No, they are not. We demonstrated this in silico, via computational chemistry approaches. Most (if not all) of the current in silico works on SARS-CoV-2 have not yet been ‘confirmed’ by experimental results.

Reviewer one: 

  • How computational values of previous published works can be considered as threshold for evaluating other new docking protocols? Are the previously reported data about enthalpy or entropy evaluation? Are the values confirmed by experimental data?

Response:  previous works, as our own, report energies of binding in the most straight forward way, as a difference of energies between the complex and the initial energies of ligand and protein. I am not sure however, what do you mean by ‘new docking protocols’ and their ‘evaluation’. As in the previous comment, experimental data have not been used yet to confirm as a one-to-one correlation with in silico predictions

Reviewer one: 

  • How much significant are the differences between different binding energies reported in table 1a? Did authors run a statistical significance analysis of these data? if yes, please provide them.

Response: we haven’t run a statistical significance analysis of our reported data for two reasons: a- no similar works have done so, so we wouldn’t provide another comparison parameter between ours and previous works (all comparisons are on the basis of previous comparisons: energetic and structural), and b- the binding efficiency of a group of ligands is an exponential function of the binding energy (small differences lead to large differences of occupancy)

Reviewer 2 Report

General comment:

This manuscript, entitled “Metal-bound Methisazone; Novel Drugs Targeting Prophylaxis and Treatment of SARS-CoV-2, a Molecular Docking Study,” authored by Ahmed et al., reports the potential application of metal bound methisazone in protection against SARS-CoV-2. This manuscript is an example of using in silico approach to probe modified drug-protein interaction. The results reported representing a potential of these drug-protein interactions. In the current Covid situation, the outcome work of this manuscript will be helpful. In my opinion, this is a good paper, and the manuscript is suitable for publication in Int. J. Mol. Sci. after the authors have addressed the following comments and questions:

Specific comments:

  • Line 89…HyperChem 8.0 (HyperCube Inc.), was used to compare potential differences between the two pathways….which pathways author is talking about here? And how hyperchem 8.0 is used for pathway identification…make it clear here.
  • Out of several metals, is there any specific reason for choosing these metals? There are at least 11 metals in the biological system available. Many others are toxic still used in pharmaceuticals at trace concentration.
  • Page number 4 - Section 2.3. Drug Candidate Analysis – Autodock itself giving you the energy of binding. Why were you using Hyperchem to calculate binding energies? Please elaborate.
  • Line 144 and table 1…..The lowest binding energy for methisazone was found with 6Y2E_A and 6W9C…..Is it true? I can see 6VYB has the lowest binding energy (-7.7) for Methisazone.
  • Line 146….how binding energy in the case of 6W9C has -1.8 kcal/mol as a cumulative increase? Several values showing here is not matching with the table or missing connecting explanations.
  • Page number 5 - Section 3.2. Docking Visualization: Just a suggestion- is there a better way to visualize structure-ligand here. Please make the figure consistent. I am sure there is the possibility to enhance those. Please describe the interacting residues in the pocket. In fig 6 and 7 those are not clear, and color selection is low. Keep important reduces to show the interaction and more informative.
  • How will authors eliminate nonspecific interaction of the drug with proteins? Whether the author has performed any negative control to verify nonspecificity?
  • Metal bound methisazone – How is the metal bonded with the compound?
  • Please make the Conclusion part a bit more attractive as it is losing the manuscript's impression at the end.

Author Response

Reviewer 2:

Comments and Suggestions for Authors

General comment:

This manuscript, entitled “Metal-bound Methisazone; Novel Drugs Targeting Prophylaxis and Treatment of SARS-CoV-2, a Molecular Docking Study,” authored by Ahmed et al., reports the potential application of metal bound methisazone in protection against SARS-CoV-2. This manuscript is an example of using in silico approach to probe modified drug-protein interaction. The results reported representing a potential of these drug-protein interactions. In the current Covid situation, the outcome work of this manuscript will be helpful. In my opinion, this is a good paper, and the manuscript is suitable for publication in Int. J. Mol. Sci. after the authors have addressed the following comments and questions:

Reviewer 2: 

  • Line 89…HyperChem 8.0 (HyperCube Inc.), was used to compare potential differences between the two pathways….which pathways author is talking about here? And how hyperchem 8.0 is used for pathway identification…make it clear here.

Response:

Correct, this point is not as clear as it should be. The ‘two different approaches’ are the geometry optimization using the different abilities of the two packages. HyperChem is a complete optimization tool, that comprises all methods (Ab Initio, DFT, etc) at the highest level (6-31G** for DFT) and it is ideal for small molecules (such as MSZ) optimization. On the other hand, Chimera offers optimization capabilities, but not at the level of accuracy that HyperChem does. Thus, since the final docking would take place in Chimera, we needed to confirm that Chimera’s model would be the same. It was not, and this is why we employed HyperChem based small structure optimizations. Those optimized structures were then used in Chimera’s based molecular docking. I erased the initial phrasing with a clearer one:

Line 91: “ The structure of Methisazone was prepared in HyperChem 8.0 (HyperCube Inc.). The metal complexes of Methisazone were also prepared in the HyperChem 8.0 environment since HyperChem offers a more powerful environment for the optimization of small molecular systems via Ab Initio and Density Functional Theory calculations than Chimera.”

Reviewer 2:

  • Out of several metals, is there any specific reason for choosing these metals? There are at least 11 metals in the biological system available. Many others are toxic still used in pharmaceuticals at trace concentration.

Response:

Our choice was based on two factors:

  • Existing data in metal- bound ligands for other viruses
  • Autodock doesn’t include parameters for some of other metals of interest, thus they were excluded from this work
  •  
  • Reviewer 2:
  • Page number 4 - Section 2.3. Drug Candidate Analysis – Autodock itself giving you the energy of binding. Why were you using Hyperchem to calculate binding energies? Please elaborate.

Response:

For comparison reasons, as described in a previous comment. We know that Chimera is best and faster for the larger systems, and we used these results in our findings. We can eliminate this point; it is probably of no use to the reader, text has been updated as follows:

Line 153 “The results per candidate complex were evaluated. AutoDock software reports the binding energies for each Methisazone-protein pair in kcal/ mol. Binding energies between the proteins and complexes are calculated as the difference between the final docked structure (protein-ligand) and the initial energies of protein and Methisazone- Metal complex.”

Reviewer 2:

  • Line 144 and table 1…..The lowest binding energy for methisazone was found with 6Y2E_A and 6W9C…..Is it true? I can see 6VYB has the lowest binding energy (-7.7) for Methisazone.
  • Response: lowest binding energy (weaker interaction). Table values are correct. -7.7 is the highest binding energy (in absolute value). Lowest and highest get often confused, depending on if you interpret as ‘value’ or ‘meaning’. I tried to clarify this by adding the ‘absolute value’:

Line 165: “The lowest binding energy as absolute value (weaker binding)”

Reviewer 2:

  • Line 146….how binding energy in the case of 6W9C has -1.8 kcal/mol as a cumulative increase? Several values showing here is not matching with the table or missing connecting explanations.

Response: The cumulative increase of 6W9C with the five metals is the sum of -0.1, -0.6, -0.5, -0.3, and -0.3. Also, the rest of the values are correct.

Reviewer 2:

  • Page number 5 - Section 3.2. Docking Visualization: Just a suggestion- is there a better way to visualize structure-ligand here. Please make the figure consistent. I am sure there is the possibility to enhance those. Please describe the interacting residues in the pocket. In fig 6 and 7 those are not clear, and color selection is low. Keep important reduces to show the interaction and more informative.

Response:

I agree. We have tried to exhibit those interaction more clearly, but the density of the structures made that impossible.

Reviewer 2:

  • How will authors eliminate nonspecific interaction of the drug with proteins? Whether the author has performed any negative control to verify nonspecificity?

Response:

: Non specific interactions were not eliminated due to the following reasons:

  • The visual results confirm the bonding to specific sites and pockets, and thus no significant distribution was observed. This observation confirms that the non specific interactions and affinities are negligible
  • Again, trying to produce comparable results to similar published works, we didn’t identify any previous publication (in the topic of molecular docking of ligands to SARS- CoV-2 proteins) that investigated non- specific interactions

Reviewer 2:

  • Metal bound methisazone – How is the metal bonded with the compound?

Response:

We have worked on detailed MSZ- Metal interactions on the highest level of theory possible. The Metal- MSZ distances suggest that the interactions are due to attractions (London, dispersion, electrostatic) that provide distances in the order of 1.5- 1.9 of covalent bonds. Added txt for clarity:

Line 123: ” The structures created, along with the Metal – Methisazone distances suggest that the complexing is occurs via electrostatic and dispersion forces (distances 1.5-1.9 times the length of the respective covalent bonds). Thus, it is concluded that Methisazone – Metal complexes are stable and can be delivered at their target.”

Reviewer 2:

  • Please make the Conclusion part a bit more attractive as it is losing the manuscript's impression at the end.

Response: added text:

Line 255: “Visualization of preferred binding sites and pockets assisted in better understanding of the combined interactions of multiple protein segments, and the possible existence of non- specific interactions. The narrow binding profiles that were observed visually confirm that non- specific interactions should be negligible in metal- Methisazone docking in the four proteins studied.

The initial optimization of Methisazone and Methisazone – Metal complexes at a near ‘infinite’ basis set, allowed for the accurate description of the Methisazone- metal complexing and the prediction of the stability of the complex. Very weak interactions at that complexing stage could be the result of either an inappropriate investigation approach or real absence of interactions between Methisazone and metals. This would lead to complexes not being stable enough to be used as potential drugs. Our approach, based on the employment of a ‘near infinite basis set’ and the combined geometry optimization – Molecular Dynamics, confirmed the sufficient binding strength of Methisazone with the metals investigated. The process that was followed can be described as:

-              Methisazone and the metal were placed in random positions (at least 20 different ones)

-              A geometry optimization reached a local energetic minimum

-              A molecular dynamics simulation was employed to move the Methisazone – metal system from equilibrium

-              If the attempt was not successful, the local minimum was the global minimum, and the complex represents the real structure. All properties of the complex were fed to Chimera for the Molecular Docking investigation (end of optimization)

-              If the attempt was successful, a new geometry optimization was run to find a new energetic minimum

-              The process was continued until the global minimum was reached, and all properties were fed to Chimera for the Molecular Docking investigation (end of optimization)

Round 2

Reviewer 1 Report

Dear authors,

thanks for assessing all my requests. In this present form the manuscript has been improved. Due to the importance of the content,  it would be great if authors  consider this work as a starting point to deepen the study with experimental data in order to validate their computational methods.

Reviewer 2 Report

I got answers to all my queries.